# Feedborne *Salmonella enterica* Serovar Jerusalem Outbreak in Different Organic Poultry Flocks in Switzerland and Italy Linked to Soya Expeller

**DOI:** 10.3390/microorganisms9071367

**Published:** 2021-06-23

**Authors:** Jule Anna Horlbog, Roger Stephan, Marc J. A. Stevens, Gudrun Overesch, Sonja Kittl, Maira Napoleoni, Valentina Silenzi, Magdalena Nüesch-Inderbinen, Sarah Albini

**Affiliations:** 1National Reference Center for Enteropathogenic Bacteria and Listeria (NENT), Institute for Food Safety and Hygiene, Vetsuisse Faculty, University of Zurich, 8057 Zurich, Switzerland; magdalena.nueesch-inderbinen@uzh.ch; 2Institute for Food Safety and Hygiene, Vetsuisse Faculty, University of Zurich, 8057 Zurich, Switzerland; stephanr@fsafety.uzh.ch (R.S.); marc.stevens@uzh.ch (M.J.A.S.); 3Center for Zoonoses, Animal Bacterial Diseases and Antimicrobial Resistance (ZOBA), Institute for Veterinary Bacteriology, Vetsuisse Faculty, University of Berne, 3012 Berne, Switzerland; gudrun.overesch@vetsuisse.unibe.ch (G.O.); sonja.kittl@vetsuisse.unibe.ch (S.K.); 4Istituto Zooprofilattico Sperimentale Umbria e Marche, Sezione di Tolentino, 62029 Tolentino, Italy; m.napoleoni@izsum.it (M.N.); v.silenzi@izsum.it (V.S.); 5National Reference Center for Poultry and Rabbit Diseases (NRGK), Institute for Food Safety and Hygiene, Vetsuisse Faculty, University of Zurich, 8057 Zurich, Switzerland; salbini@vetbakt.uzh.ch

**Keywords:** *Salmonella*, *S.* Jerusalem, poultry, organic, feed, soya, WGS, cgMLST, outbreak

## Abstract

Poultry feed is a leading source of *Salmonella* infection in poultry. In Switzerland, heat-treated feed is used to reduce *Salmonella* incursions into flocks in conventional poultry production. By contrast, organic feed is only treated with organic acids. In 2019, the Swiss National Reference Center for Enteropathogenic Bacteria identified the rare serovar *S.* Jerusalem from samples of organic soya feed. Further, in July 2020, the European Union’s Rapid Alert System for Food and Feed published a notification of the detection of *S.* Jerusalem in soya expeller from Italy. During 2020, seven *S.* Jerusalem isolates from seven different poultry productions distributed over six cantons in Switzerland were reported, providing further evidence of a possible outbreak. Using whole-genome sequencing (WGS), *S.* Jerusalem isolates from feed and from animals in Switzerland were further characterized and compared to *S.* Jerusalem from organic poultry farm environments in Italy. WGS results showed that feed isolates and isolates from Swiss and Italian poultry flocks belonged to the sequence type (ST)1028, grouped in a very tight cluster, and were closely related. This outbreak highlights the risk of spreading *Salmonella* by feed and emphasizes the need for a heat-treatment process for feed, also in organic poultry production.

## 1. Introduction

*Salmonella* are Gram-negative, rod-shaped bacteria belonging to the family of Enterobacteriaceae. They are facultatively anaerobic and form peritrichous flagella for motility. The genus *Salmonella* contains two species, *S. enterica* and *S. bongori* [1]. *S. enterica* consists of six subspecies based on genomic and biochemical modifications. The whole group is complex and includes more than 2500 different serovars [2].

As enteropathogenic agents, *Salmonella* play an important role in both human and animal health [3]. The main reservoirs of *Salmonella* are the gastro-intestinal tracts of healthy farm animals and humans, but *Salmonella* are also widely distributed in the environment [4]. The European Food Safety Authority (EFSA) states ‘eggs and egg products’ are the most implicated food vehicle in foodborne *Salmonella* outbreaks [3]. This explains and justifies the various efforts of authorities and producers to implement national control programs in poultry.

In Switzerland, a voluntary control program for *Salmonella* (*S*). serovar Enteritidis was introduced in mid-1993 by the poultry integrator companies [5]. As of the beginning of 1994, it was made a mandatory national control program to eliminate *S.* Enteritidis and *S.* Typhimurium from layer flocks and *S.* Enteritidis, *S.* Typhimurium, *S.* Hadar, *S.* Infantis, and *S.* Virchow from parent flocks [6]. Switzerland does not have grandparent flocks; thus, imported layer and broiler parents are tested to be *Salmonella*-free, and a stringent inland control program is in place for hatcheries, parents, and layers, and since 2008 also for broiler and turkey flocks [6]. This control program has proven successful. Flocks testing positive for one of the above mentioned serovars are culled. Recovery of other serovars, which are not isolated from internal organs other than the intestinal tract, are not regulated. However, it is recommended to slaughter such broiler flocks at the end of the line and to thoroughly clean and disinfect the stables [7].

Other than *Salmonella*-positive animals, feed is a well-recognized leading source of *Salmonella* infections in poultry [8,9,10]. During harvesting, within the production process at the mill, or during storage, the feed can be contaminated. Young chicks without fully developed, protective gut microbiota are especially susceptible to *Salmonella* colonialization via feed and the required infectious dose has been proven to be very low [11,12]. Through the shedding of feces, the farm surroundings, such as the bedding, soil, water, and litter, are often identified as sources of spreading and distribution [13,14]. *Salmonella* can infect a whole flock within 2–10 days [15] and survive in the environment between flocks. Different studies have shown the direct link between feed and end-product contamination with *Salmonella*. For example, Shirota et al. found the same *S.* Enteritidis in contaminated feed as well as in egg contents [16]. *S.* Typhimurium was also found in breeder diets as well as in broilers, as described by Mac Kenzie and Bains in 1976 and Jones et al. in 1991 [17,18]. Therefore, strict control and prevention measures for *Salmonella* in feed were suggested early on. A study of Veldman et al. showed that a feed-pelleting temperature above 80 °C can reduce *Salmonella* contamination below the limit of detection [19]. In a study of Magossi et al. in the United States, every sampled feed mill had at least one culture positive sample for *Salmonella*, but the number of positive sites decreased following the line of production towards the final product [20]. In Switzerland, heat-treated feed has been used predominantly in poultry production since 1996 to reduce *Salmonella* incursions into flocks [5]. However, organic production systems are an exception because they are subject to the Swiss organic farming regulation. Organic feed, however, is not heat-treated but treated with organic acids, which may be less effective at controlling *Salmonella*, and the feed preparation techniques used must be as natural and energy-saving as possible [21].

Prior to this study, in 2019, the Swiss National Reference Center for Enteropathogenic Bacteria (NENT) identified the rare serovar *S.* Jerusalem in six isolates from samples of organic soya feed. Furthermore, in July 2020, the European Union’s Rapid Alert System for Food and Feed (RASFF) notification 2020.3066 published the detection of the same serovar (presence/25 g) in soya expeller from Italy [22]. In parallel, a poultry veterinarian became aware that *S.* Jerusalem had been identified almost at the same time, in two poultry farms in two different Swiss cantons. The observation was reported to the National Reference Centre for Poultry and Rabbit Diseases (NRGK), and an investigation of a possible novel introduction of this rare serovar was initiated. Concurrently, seven *S.* Jerusalem isolates from seven different organic poultry productions distributed over six cantons in Switzerland were reported by the Center for Zoonoses, Animal Bacterial Diseases and Antimicrobial Resistance (ZOBA), providing further evidence of a possible outbreak.

The aim of this study was to further characterize the available *S.* Jerusalem isolates from Swiss feed and poultry by whole-genome sequencing, to compare them to available Italian isolates from the poultry environment, and to identify a possible epidemiological link.

## 2. Materials and Methods

### 2.1. Bacterial Isolates

At the end of 2019, the NENT identified six *S.* Jerusalem from feed (organic soya cake). In 2020, the Center for Zoonoses, Animal Bacterial Diseases and Antimicrobial Resistance (ZOBA) identified the same serovar in seven isolates submitted for serotyping by three different diagnostic laboratories. These isolates originated from seven different poultry farms from three different regions and six different cantons of Switzerland: Northwestern Switzerland (Aargau AG, Solothurn SO), Central Switzerland (Zug ZG), and Northeastern Switzerland (Zürich ZH, Thurgau TG, St. Gallen SG).

Moreover, three *S.* Jerusalem isolates from the poultry environment from the Marche region, Italy, were provided for further characterization by the Istituto Zooprofilattico Sperimentale Umbria e Marche. In total, 16 isolates were available for analysis (Table 1).

### 2.2. Whole-Genome Sequencing Analysis and Bioinformatics

Bacteria were plated from cryocultures onto sheep blood agar (Difco Laboratories, Franklin Lakes, NJ, USA) and incubated overnight at 37 °C. Total DNA was isolated from the overnight culture using the DNeasy Blood and Tissue Kit (Qiagen, Hilden, Germany) according to the manufacturer’s protocol. Sequencing libraries were obtained using the Illumina Nextera flex DNA preparation kit (Illumina, San Diego, CA, USA) according to the manufacturer’s protocol. A final DNA concentration of 1.3–1.4 pM, depending on the run, was used for sequencing on an Illumina MiniSeq (Illumina, San Diego, CA, USA). Reads were assembled using Shovill 1.0.4 and Spades 3.13.1 [23,24], using default settings. The assembly was filtered, retaining contigs >500 bp and annotated using the NCBI prokaryotic genome annotation pipeline [25].

Sequence types (STs) were determined using a 7 house-keeping gene-based MLST scheme and Ridom SeqSphere+ software version 7.2.3 (Ridom GmbH, Münster, Germany). Sequences were blasted against a cgMLST scheme based on a 3002-locus cgMLST scheme using Ridom SeqSphere+. Complex types (CTs) were assigned upon submission to the *Salmonella* cgMLST Ridom SeqSphere+ server (https://www.cgmlst.org/ncs/schema/4792159/ accessed on 4 May 2021). A cluster was defined as a group of isolates with ≤10 different alleles between neighboring isolates. High-quality SNPs were identified by mapping reads against the draft assembly of the strain 50,722 using the CFSAN SNP pipeline [26]. A maximum-likelihood phylogenetic tree was constructed from the SNP matrix using IQ-TREE v2.0.3 [27] with the generalized time-reversible (GTR) model and gamma distribution with 100 bootstraps to assess confidence. The number of invariant sites was estimated from a core genome alignment generated with parsnp [28] and passed to IQ-TREE.

Software and databases were all updated in February 2021 and default parameters were used for all in silico analyses.

### 2.3. Antimicrobial Susceptibility Testing

Susceptibility testing to 13 antimicrobial agents was tested using the disc diffusion method according to CLSI protocols and evaluated according to CLSI criteria [29]. Briefly, each isolate was suspended in 0.8% NaCl solution and adjusted to 0.5 McFarland standard. Swaps were used to evenly spread the inoculum on Müller–Hinton agar plates (Oxoid, Hampshire, UK) before antibiotic discs were applied. After overnight incubation, inhibition zones were measured. The antibiotics tested were ampicillin, amoxicillin/clavulanic acid, cefalotin, cefotaxime, ciprofloxacin, gentamicin, tetracycline, streptomycin, chloramphenicol, kanamycin, nalidixic acid, and sulfamethoxazole/trimethoprim (Becton Dickinson, Franklin Lakes, NJ, USA). Isolates exhibiting resistance to three or more antibiotic classes (counting ß-lactams as one class) were classified as multidrug resistant.

## 3. Results

The genomes of the 16 *S.* Jerusalem isolated were completely sequenced using Illumina technology. The sequencing outputs were between 695,228 and 1,624,592 paired-end reads of 150 bp, resulting in a genome coverage of 44 to 102 times. The Illumina-reads files passed the standard quality checks using the software package FastQC 0.11.7 (Babraham Bioinformatics, Cambridge, UK), with exception for the “per base sequence content” check. This error was expected because of the transposon-based sequence libraries that were produced.

The draft genomes of the 16 *Salmonella* isolates consisted of 4,725,864 to 4,787,949 bp divided over 34–43 contigs with an N50 between 287,235 and 402,221 bp (Table 2). The GC-content of the genomes was 52.18 mol% for all strains. The genomes harbored 4377–4444 genes (Table 2), as predicted by the NCBI prokaryotic genome annotation pipeline.

All strains belonged to the MLST sequence type 1028 and grouped in a very tight cluster based on WGS. Except for strain 50772, which was found already in 2003, the isolates differed by only up to 3 cg-alleles (Figure 1).

Similarly, in a SNP-based analysis, the 15 recent isolates clustered distantly from 50,722 (Figure 2). The 15 genomes differed by 1–10 core genome SNPs, suggesting that the isolates are epidemiologically linked.

None of the isolates showed any resistances towards the tested 13 antimicrobial agents.

## 4. Discussion

In this outbreak, the investigation revealed that organic soy-based feed contaminated with *S*. Jerusalem had been in circulation in Switzerland since 2019. Following an RASFF notification published in July 2020, *S.* Jerusalem was isolated from boot socks, dust, and fecal samples from organic broiler, layer, and layer breeder farms from Switzerland and Italy.

The chain of evidence presented in this study is thus strongly suggestive of feed as the origin of flock contamination. The possibility of a pseudo-outbreak similar to one such outbreak involving boot socks contaminated with the rare serovar *S.* Goverdhan [31] was ruled out, since the isolates were originally isolated by three different diagnostic laboratories and feed, dust, and fecal samples were also found to be positive in this study.

Previously, further evidence for epidemiological links was obtained by performing time-consuming and sometimes unsatisfactory methods, such as pulsed-field gel electrophoresis (PFGE), multilocus variable-number tandem-repeat analysis (MLVA), or phage typing [32]. Nowadays, WGS analysis is increasingly replacing these analytical processes. In the present study, the link suggested by *Salmonella* serotyping was further supported by the WGS data, which clearly demonstrated the very close relatedness between the *S*. Jerusalem from the feed and those from the environments of the poultry farms. Notably, all poultry isolates originated from organic production, while no *S.* Jerusalem isolates were recovered from conventional feed for poultry production. In conventional production, expanded (heat-treated) feed has been used in Switzerland since 1996 [5]. Organic feed, however, is not heat treated but treated with organic acids, which may be less effective at controlling *Salmonella*.

Serotyping of all *Salmonella* isolates from both feed and farm samples is standard procedure in Switzerland. However, feed isolates and farm isolates are monitored by different authorities: data from feed isolates are made available to Agroscope, Federal Department of Economic Affairs, Education and Research, while those from the farm, animal, and human isolates are reported to the Federal Food Safety and Veterinary Office, Federal Department of Home Affairs. The incentive to investigate the herein described *S.* Jerusalem cases came from a poultry veterinarian noticing two *Salmonella* Jerusalem cases within her client stocks. Monitoring of both feed and farm *Salmonella* isolates by a single authority would thus be preferable to identify feed-to-food-associated risks. A future system to oversee the *Salmonella* serotyping data from a “from feed to food” perspective, including RASFF notifications, would be desirable.

Whole-genome sequencing played a key role in showing the close relatedness between the isolates from the different poultry flocks in Switzerland and Italy, and linking them to the feed isolates. This outbreak highlights the risk of spreading *Salmonella* by feed and emphasizes the need for enhanced *Salmonella* prevention measures, such as a heat-treatment process, also for feed in organic poultry production.

## Figures and Tables

**Figure 1 microorganisms-09-01367-f001:**
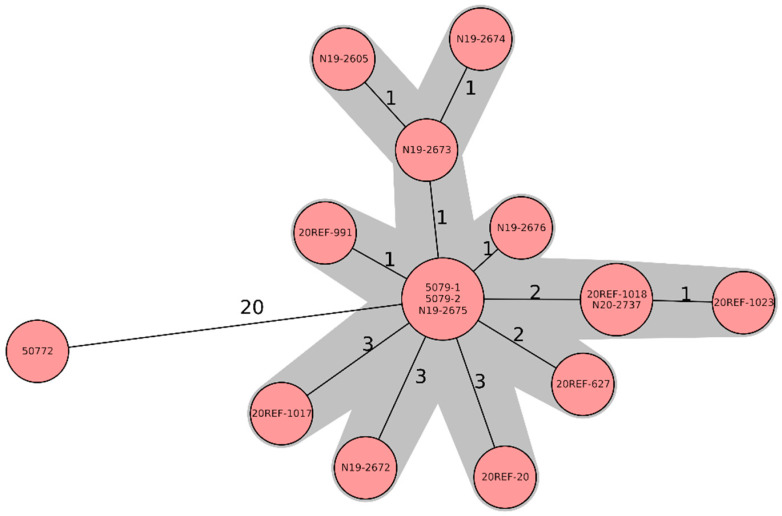
Minimum-spanning tree based on cgMLST allelic profiles of 16 *Salmonella* Jerusalem isolates, six feed isolates (soya), and ten chicken isolates. Each circle represents an allelic profile based on a sequence analysis of 3002 cgMLST target genes. The numbers on the connecting lines represent the number of allelic differences between two strains. Each circle contains the strain ID(s), as listed in Table 1.

**Figure 2 microorganisms-09-01367-f002:**
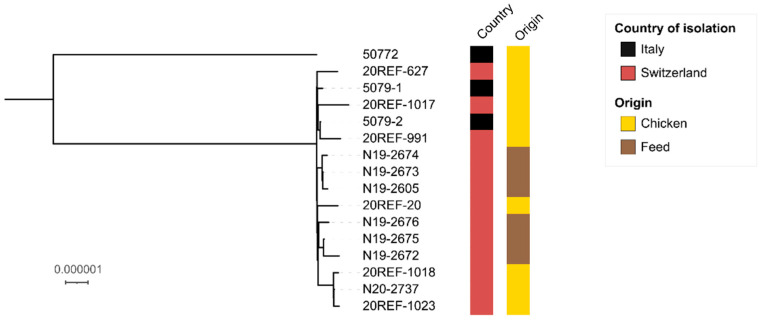
Maximum-likelihood phylogenetic tree of 16 *Salmonella* Jerusalem isolates. The country and origin of isolation are labelled according to the legend. The scale bar indicates the number of variant sites in a 4.381 Mbp core genome. The tree was visualized using iTOL [30].

**Table 1 microorganisms-09-01367-t001:** Origin of the *Salmonella* Jerusalem isolates.

Strain ID	Sample Type	Host	Production Type	Date of Isolation	Swiss Canton/Italian Region
N19-2605	soya cake	n.a.	organic	30 October 2019	n.a.
N19-2672	soya cake	n.a.	organic	5 November 2019	n.a.
N19-2673	soya cake	n.a.	organic	5 November 2019	n.a.
N19-2674	soya cake	n.a.	organic	5 November 2019	n.a.
N19-2675	soya cake	n.a.	organic	5 November 2019	n.a.
N19-2676	soya cake	n.a.	organic	5 November 2019	n.a.
N20-2737	Feces	layer breeder	organic	21 December 2020	ZH
20REF-20	boot sock	laying hen	organic	12 January 2020	ZH
20REF-627	boot sock	laying hen	organic	29 September 2020	SG
20REF-991	Dust	laying hen	organic	10 November 2020	SO
20REF-1017	boot sock	laying hen	organic	16 December 2020	ZG
20REF-1018	Dust	laying hen	organic	15 December 2020	AG
20REF-1023	boot sock	laying hen	organic	18 December 2020	AG
5079_1	boot sock	broiler	organic	25 January 2021	Marche
5079_2	boot sock	broiler	organic	25 January 2021	Marche
50772	poultry manure from composting plant	poultry	unknown	3 November 2003	Marche

Swiss cantons: AG, Aargau; SG, St. Gallen; SO, Solothurn; ZG, Zug; ZH, Zürich; Italian region: Marche; n.a., not applicable.

**Table 2 microorganisms-09-01367-t002:** Characteristics of the sequences of the 16 *Salmonella* Jerusalem isolates.

Strain	Year of Isolation	Origin	N_reads_ ^a^	Coverage	Length (bp)	Contigs	N50	L50	N Genes	Accession No.
N19-2605	2019	Feed	966,373	61	4,782,067	40	334,862	5	4435	JAFJZF000000000
N19-2672	2019	Feed	887,841	56	4,781,041	35	402,221	4	4432	JAFJZG000000000
N19-2673	2019	Feed	957,477	61	4,780,566	34	348,029	5	4434	JAFJZH000000000
N19-2674	2019	Feed	982,883	62	4,780,191	41	347,991	6	4437	JAFJZI000000000
N19-2675	2019	Feed	992,783	63	4,781,953	36	348,407	5	4432	JAFJZJ000000000
N19-2676	2019	Feed	802,128	51	4,779,956	34	347,991	5	4433	JAFJZK000000000
N20-2737	2019	Chicken	1,070,707	68	4,787,949	40	334,862	5	4439	JAFJZL000000000
20REF-20	2020	Chicken	891,287	56	4,781,073	38	348,023	5	4434	JAFJZC000000000
20REF-627	2020	Chicken	695,228	44	4,778,515	40	348,023	6	4435	JAFJZD000000000
20REF-991	2020	Chicken	919,591	58	4,779,174	43	325,448	6	4438	JAFJZE000000000
20REF-1017	2020	Chicken	1,624,592	102	4,774,988	40	300,861	6	4435	JAFJYZ000000000
20REF-1018	2020	Chicken	1,263,023	80	4,780,243	36	348,254	5	4434	JAFJZA000000000
20REF-1023	2020	Chicken	1,187,079	75	4,780,375	38	348,393	5	4437	JAFJYY000000000
5079-1	2021	Chicken	921,944	58	4,779,660	36	347,885	5	4430	JAGYFQ000000000
5079-2	2021	Chicken	1,024,829	64	4,773,619	39	347,983	6	4434	JAGYFP000000000
50772	2003	Chicken	907,688	58	4,725,864	39	287,235	6	4377	JAGYFN000000000

^a^ Refers to the number of reads in one set of the paired-end reads of 150 bp.

## Data Availability

This Whole Genome Shotgun project has been deposited at DDBJ/ENA/GenBank under the accession no. JAFJYY000000000 to JAFJZA000000000, JAFJZC000000000 to JAFJZL000000000 and JAGYFN000000000, JAGYFP000000000, JAGYFQ000000000. The versions described in this paper are JAFJYY000000000 to JAFJZA000000000, JAFJZC000000000 to JAFJZL000000000 and JAGYFN000000000, JAGYFP000000000, JAGYFQ000000000. The raw sequencing reads have been deposited in the SRA under the bioproject PRJNA705077.

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
