# Peer review of "Feedborne Salmonella enterica Serovar Jerusalem Outbreak in Different Organic Poultry Flocks in Switzerland and Italy Linked to Soya Expeller"

_microorganisms, 2021, doi:10.3390/microorganisms9071367_

Round 1

Reviewer 1 Report

This is an interesting article that examines and characterizes the available S. Jerusalem isolates from Swiss feed and poultry by whole genome sequencing. In addition, the research compares isolates to the available Italian ones from the poultry environment, to identify a possible epidemiological link. The topic is current since Salmonella is one of the main causes of foodborne disease worldwide. The paper is well written, and the results should be useful for the scientific community. For all these reasons, this paper is, in my opinion, suited for publication. Moreover, there are some needed minor revisions: 

L. 43: Please add a reference about the Report EFSA that you cited

 L. 116: Please add the reference “Detection of Campylobacter from poultry carcass skin samples at slaughter in southern Italy, Pepe et al., 2009” about methods

Author Response

Dear Reviewer,

Thank you for your input. Points raised are adressed point-by-point in the following.

L.43: Please add a reference about the Report EFSA that you cited

This is a good point – thank you. The reference was added. (3)

L.116: Please add the reference “Detection of Campylobacter from poultry carcass skin samples at slaughter in southern Italy, Pepe et al., 2009” about methods

The suggested reference describes the investigation of skin samples at slaughterhouse levels for Campylobacter and uses multiplex PCR for species identification. The Salmonella strains originate from boot socks or fecal or feed samples; neither the sampling method, nor the following strain isolation follows the same methods as described in the referred paper. We did not use any multiplex PCR for species identification. Moreover, in the Swiss legislation the sampling on primary production (samples that were taken in this study) and on slaughterhouse level (as in the suggested reference) are regulated in different places and on different levels. Therefore, the reference was not added.

Reviewer 2 Report

The work is interesting and relevant because control programs for specific Salmonella serotypes may lead to the expansion of other serotypes. However, the authors can obtain more information from the analyses performed, as indicated in the following comments.

Line 24: Delete line break.

Line 117: The authors should add more information about the kits used to prepare the libraries, the amount of DNA used…

Line 126: The authors could analyze the genomic presence of resistance genes using software such as ResFinder.

Line 132: Briefly describe the method.

Line 156: Perhaps the authors could add a figure comparing the genomes of the bacteria using programs such as BLAST Ring Image Generator (BRIG).

Line 156: I think the authors could take more advantage of the data obtained from whole genome sequencing. There is limited genomic analysis of the strains, and this makes the work too narrow. For example, they could analyze the presence of pathogenicity islands and virulence genes. This would give important information on the pathogenic capacity of the strains of this serotype. The distribution of this pathogen is as interesting as its pathogenic capacity.

Author Response

Dear Reviewer,

Thank you for your input. Points raised are adressed point-by-point in the following.

Line 24: Delete line break.

Thank you for pointing this out. The line break is now deleted.

Line 117: The authors should add more information about the kits used to prepare the libraries, the amount of DNA used…

The kit used for library preparation and information about the amount of DNA used for sequencing is now added in L31-L38.

Line 126: The authors could analyze the genomic presence of resistance genes using software such as ResFinder.

The reviewer is right. However, this was not the aim of this study. We integrated the phenotypic resistance data as further characterization data. No resistances against any antibiotics used were found. However, the genomes are all made publicly accessible and if a genetic screening for resistance genes would be interesting for anyone, this analysis could be done.

Line 132: Briefly describe the method.

In L170-L174 a few sentences about the method have been added.

Line 156: Perhaps the authors could add a figure comparing the genomes of the bacteria using programs such as BLAST Ring Image Generator (BRIG).

Thank you for rising this and also the next comment, to add more value to the obtained WGS data. The focus of this study was to establish an epidemiological link between the feed isolates and the farm cases. Our intention was to replace the rather time consuming and often unsatisfactory methods of for example PFGE and MLVA by WGS analysis to proof the epidemiological link. We have added a sentence L232-L235 to clarify this aspect. However, we do agree to the point raised and have tried to address it by performing an additional SNP analysis to obtain a deeper comparison of the strains and add value to the WGS data. Figure 2 has been added to the manuscript and also a relevant part in the methods (L 158-L164) and the results (L196-L198) section.

Line 156: I think the authors could take more advantage of the data obtained from whole genome sequencing. There is limited genomic analysis of the strains, and this makes the work too narrow. For example, they could analyze the presence of pathogenicity islands and virulence genes. This would give important information on the pathogenic capacity of the strains of this serotype. The distribution of this pathogen is as interesting as its pathogenic capacity.

This was not the aim of this study. Nevertheless, genomes all made publicly available and if the question arises this analysis can be performed by anyone interested.

Reviewer 3 Report

Non-typhoidal Salmonella serovars cause self-limiting gastroenteritis in humans and animals. However, it can also cause life-endangering systemic illness in immunocompromised hosts, e.g., preterm infants and elders. Salmonella occurrence in poultry flocks and livestock herds decreases the rentability of farming. Horlbog et al. submitted their manuscript entitled „Feedborne Salmonella enterica serovar Jerusalem outbreak in different organic poultry flocks in Switzerland and Italy linked to soya expeller“ to Microorganisms. This manuscript deals with occurrence of rare S. Jerusalem, mainly in Switzerland.

The manuscript is interesting and conscientiously written. Thus, I have only a few notices or recommendation:

L2: Feedborne of Foodborne?

L61: I recommend replacing the term microflora with microbiota.

L99-103: SIX S. Jerusalem from SEVEN farms??

I recommend requalifying the manuscript from Article to Communication (character of the work, low number of the used methods and results, length of the manuscript, low number of references).

Author Response

Dear Reviewer,

thank you for your input. The points raised are addressed one-by-one in the following.

L2: Feedborne of Foodborne?

We would suggest staying with the title as feedborne. The soy expeller is used to feed the chicken and therefor the epidemiological link, which is the focus of this study. In our understanding foodborne would refer to foodstuff used for human consumption rather than for animals. We are aware that the term does not comply with the best English grammar, nevertheless it is commonly used in the field and therefore widely accepted in use as common sense.

L61: I recommend replacing the term microflora with microbiota.

Done. Now L66.

L99-103: SIX S. Jerusalem from SEVEN farms??

From Switzerland we have six S. Jerusalem isolated from feed and seven isolates from chickens from seven different farms. We have rephrased the section. (L120-L126)

I recommend requalifying the manuscript from Article to Communication (character of the work, low number of the used methods and results, length of the manuscript, low number of references).

We were not sure if communications are considered for the special issue “Salmonella and Samlonellosis” but would comply with the editor's preferences if this is a possibility.

Round 2

Reviewer 2 Report

I encourage the authors to continue working with these strains, performing both phenotypic and genotypic analyses to determine the resistance of these strains in feed.